

# Comparing the catch composition, profitability and discard survival from different trammel net designs targeting common spiny lobster (*Palinurus elephas*) in a Mediterranean fishery

Gaetano Catanese[1,2], Hilmar Hinz[3], Maria del Mar Gil[1,3], Miquel Palmer[3], Michael Breen[4], Antoni Mira[5], Elena Pastor[1], Amalia Grau[1], Andrea Campos-Candela[3], Elka Koleva[3], Antoni Maria Grau[5] and Beatriz Morales-Nin[3]

[1] Laboratori d'Investigacions Marines i Aqüicultura (LIMIA), Govern de les Illes Balears, Port d'Andratx, SPAIN
[2] Instituto de Investigaciones Agroambientales y de Economía del Agua (INAGEA), Palma de Mallorca, SPAIN
[3] Instituto Mediterraneo de Estudios Avanzados (IMEDEA; CSIC-UIB), Esporles, Illes Balears, SPAIN
[4] Institute of Marine Research (IMR), Bergen, NORWAY
[5] Direcció General de Pesca i Medi Marí, Govern de les Illes Balears, Palma de Mallorca, SPAIN

Corresponding author
Gaetano Catanese,
gcatanese@dgpesca.caib.es

## ABSTRACT

In the Balearic Islands, different trammel net designs have been adopted to promote fisheries sustainability and reduce discards. Here, we compare the catch performance of three trammel net designs targeting the spiny lobster *Palinurus elephas* in terms of biomass, species composition and revenue from commercial catches and discards. Designs differ in the netting fiber type (standard polyfilament, PMF, or a new polyethylene multi-monofilament, MMF) and the use of a guarding net or *greca*, a mesh piece intended to reduce discards. Catches were surveyed by an on-board observer from 1,550 netting walls corresponding to 70 nets. The number of marketable species captured indicated that the lobster trammel net fishery has multiple targets, which contribute significantly to the total revenue. The discarded species ranged from habitat-forming species to elasmobranches, but the magnitude of gear-habitat interactions on the long term dynamics of benthos remains unclear. No relevant differences in revenue and weight of discards were detected after Bayesian analyses. However, the species composition of discards was different when using *greca*. Interestingly, high immediate survival was found for discarded undersized lobsters, while a seven day survival assessment, using captive observation, gave an asymptotic estimate of survival probability as 0.64 (95% CI [0.54–0.76]). Therefore, it is recommended that it would be beneficial for this stock if an exemption from the EU landing obligation regulation was sought for undersized lobsters in the Balearic trammel net fishery.

## INTRODUCTION

The discarding of the unwanted or non-marketable fraction of catch has globally been recognized as one of the most pressing issues for fisheries management (*Hall & Mainprize, 2005*). Discarding is seen as a wasteful non-usage of marine natural resources that can negatively affect the sustainable exploitation of marine biological resources and marine ecosystems, as well as the long term financial viability of fisheries (*Kelleher, 2005*). For this reason, the EU Common Fisheries Policy (CFP) introduced the landing obligation (LO) for such discards (Regulation EC 1380/2013, *European Union (EU)*). For Mediterranean fisheries, the new regulation mainly affects the discarding of juveniles of species for which a minimum landing size is mandatory. The goal of the regulation is to provide incentives for more selective fishing and to provide reliable data to facilitate better recording control over actual catches.

Not all fisheries are equally affected by the European LO. In this context, objective knowledge of discard composition related to specific fisheries is needed to provide evidence for scientific advice to implement adequate management policies and mitigation methods if required. Data on the discard amount and composition are widely available for the large, closely managed, trawl fisheries in northern Europe and the Mediterranean, while for small-scale fisheries, particularly from the Mediterranean, such data are still sparse (*Tsagarakis, Palialexis & Vassilopoulou, 2014*). However, in the Mediterranean, small-scale fisheries play an important socioeconomic role and have a long-lasting tradition (*Morales-Nin, Grau & Palmer, 2010*; *Stergiou et al., 2006*). Small-scale fleets represent 80% (42,000 boats) of the EU Mediterranean fishing vessels and contribute to 12% of the EU catches (*Maynou et al., 2013*; *Morales-Nin, Grau & Palmer, 2010*). Current reported discard rates of trammel net fisheries in the Mediterranean vary between 10% and 43% (*Tsagarakis, Palialexis & Vassilopoulou, 2014*) depending on the species targeted. Discarding can be highly variable throughout time and depend not only on the catch composition but also on market conditions, and legal restrictions (*Batista, Teixeira & Cabral, 2009*; *Veiga et al., 2016*). While most of the existing studies report on the catch and bycatch of vertebrate species few studies to date have include invertebrate or habitat-forming organisms in their assessment.

Most of the small-scale vessels in Majorca (Balearic Islands) use trammel nets of varying designs to target both fish and shellfish (*Quetglas et al., 2016*; *Palmer et al., 2017*). The spiny lobster, *P. elephas*, is one of the economically most prized species affected by the LO in Balearic Islands and is listed in Annex III of the Regulation EC 1967/2006 (December 21, 2006) as having a minimum size (see below). However, the LO regulations contrast substantially with the management rules currently applied by the local government, which requires releasing undersized juveniles and ovigerous (egg-bearing) female lobsters back to the sea. To-date, local management rules include: (a) open fishing season limited to the time period between 1st April and 31st August to avoid the breeding period; (b) a minimum landing size of 240 mm of total length (90 mm of carapace length, CL), which approximately coincides with the size at first maturity (*Goñi & Latrouite, 2005*); (c) capture and retention on board and commercialization of ovigerous
females is prohibited at any age and size; (d) soaking time of the nets cannot exceed 48 h to minimize discard mortality; and (e) the mesh size (full mesh knot to knot min. 133 mm) and the total length of trammel nets (5,000 m) per vessel are also regulated (*BOE, 11324/2001*; BOIB, 38/2001, *Ministerio de Agricultura, Pesca y Alimentación*). In general fishing boats in the Balearic Islands operate two trammel nets per fishing trip composed of 25–28 panels with and approximate length of 100 m to reach the allowed maximum length (i.e., two 2.5 km nets). Fishers generally use mesh sizes considering full mesh knot to knot of 133–160 mm. *P. elephas* is generally targeted at depth between 50 and 150 m. In the Balearic Islands approximately 110 fishing vessels target spiny lobster at some moment of the year which comprises about 42.6% of the total artisanal small-scale fishery in the Balearic Islands (Data of Balearic Government, 2017, unpublished data).

Fishing with trammel nets for spiny lobster has been one of the most important fishing activities of the artisanal sector in Majorca for the 2004–2015 period in terms of effort (21.9%; 3204.1 fishing trips/year), landings (13.8%; 58.4 metric tonnes/year) and revenue (25.7% gross revenue; 999674.00 Euros/year) (*Palmer et al., 2017*). These figures include both the target species and the commercialized bycatch (*Palmer et al., 2017*). The annual spiny lobster catches landed at the central wharf in Majorca for all the commercial small-scale vessels over the last 15 years (*OPMallorcaMar*, 2002–2017, unpublished data) ranged between 9.6 and 17.6 tonnes/year (mean = 13.4 tonnes/year), and the average first sale price ranged between 36.0 and 58.5 Euros/kg (mean = 43.4 Euros/kg).

Due to its prominence, the general characteristics of the Majorcan lobster trammel net fishery are relatively well known (*Iglesias et al., 1994*; *Quetglas et al., 2004*; *Merino et al., 2008*). Furthermore, fishermen themselves want to ensure the sustainability of this fishery by reducing any potential negative impacts (*Amengual-Ramis et al., 2016*). Accordingly, they have begun to introduce changes in the fishing tactics, but the effectiveness of these have not yet been evaluated. For example, exchanging the standard polyfilament (PMF) nets for a new polyethylene multi-monofilament (MMF) net combined with the use of a special design (*greca*), also referred to as a selvedge or guarding net, which is intended to reduce the discards from the sea bottom. While monofilament netting are increasingly used the adoption of a guarding net has only been trialed by fishers that participated in the present study. Therefore, one aim of the current study was to assess the effectiveness of these technological adaptations that fishermen are beginning to implement in the Mediterranean to reduce discards, and to provide a detailed description of catches and revenue from the wanted and unwanted fractions. Additionally, we investigate the immediate survival of the most frequently discarded species, including undersized spiny lobsters. Furthermore, we estimate the "short-term" (seven day) survival of undersized spiny lobsters by means of observations in captivity.

# MATERIALS AND METHODS

## Sampling

A total of 35 fishing trips on three, small, commercial boats with lobster trammel nets were surveyed in Majorcan waters off the Port d'Andratx and Portopetro harbors from

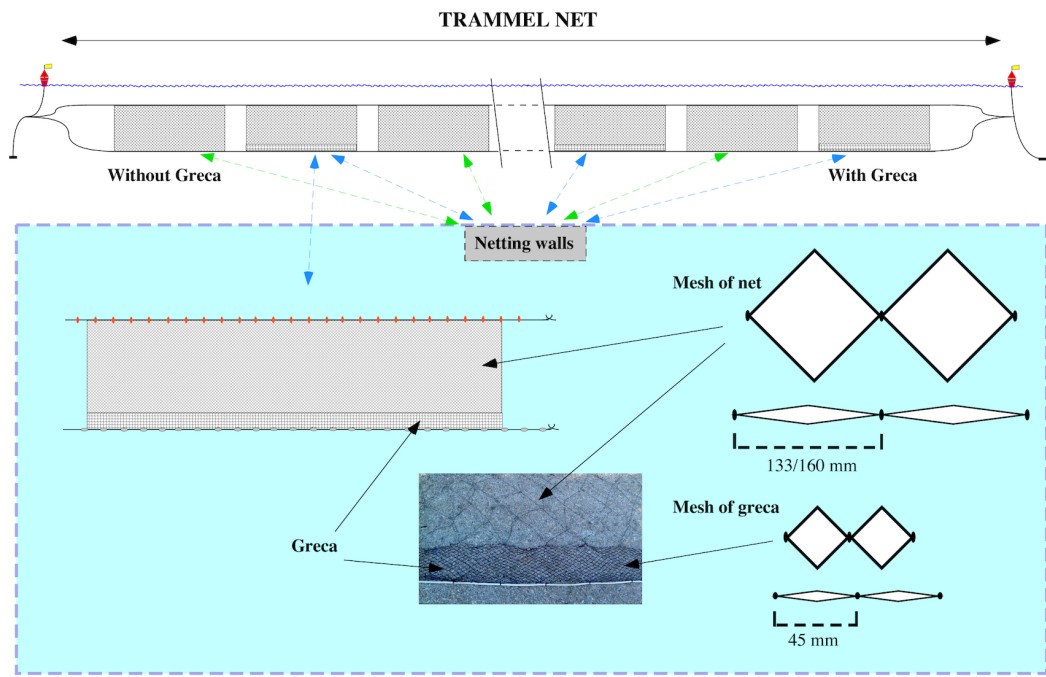

**Figure 1 Details of the strip added at the bottom of trammel nets (*greca*).** *Greca* net approximately 20 cm of nylon material with a mesh size of 45 mm that is sown to the bottom of the main net along its entire length.                                                 

April to August in 2015 and 2016. Scientific sampling was carried out on fishing boats that conducted their normal commercial activity while allowing for onboard observers under the condition of minimal disruption to their activity. Sampling protocols were therefore adopted to maximizing data collection while mitigating disturbance of fishing procedures. Each boat was rigged with two sets of trammel nets. A trammel net set consisted of several (between 10 and 30) netting walls (or "netting panels"), each approximately 100 m long and with a mesh size of 160 mm (full mesh knot to knot). Two manufacturing materials are currently used by fishermen in the Balearic Islands. Netting walls studied were either composed of standard PMF or MMF. A total of 981 PMF and 499 MMF netting walls were sampled. Trammel nets were either composed of one or a combination of the two materials.

Additionally, conventional MMF nets were compared to MMF nets with an additional modification called *greca* (Fig. 1), which is considered to reduce the capture of unwanted benthic organisms and reduce negative impacts on demersal communities. A *greca* is a piece of net approximately 20 cm high made of a thicker nylon material with a mesh size of 45 mm that is sown to the bottom of the main net along its entire length. Trammel nets to test the effect of a *greca* were composed of 10 netting walls alternating between five MMF with a *greca* and five MMF without a *greca*. Data from a total of 14 trammel net deployments were recorded. In total, 70 netting walls with *greca* and 70 without were evaluated. Apart from the netting materials and the *greca*, all trammel nets were comparable in their setup, i.e. mesh size, net length, floats and lead weighted rope used.

The date and time of deployment, location, soaking time, depth and type of net (PMF, MMF or MMF with *greca*) were recorded by an observer on board. The organisms caught were categorized into wanted ("marketable") and unwanted catches ("non-marketable") by the observer following the decisions made by the fishermen about the catch. All the marketable species and most of the large discarded species can be assigned to a specific netting wall because they remain entangled in the net. Thus, the netting wall was considered as the sample unit for those animals, which were immediately identified to the lowest taxonomic level possible, which was usually the species level, and measured with a ruler or (if not possible) visually inspected and assigned into 10 cm length classes. Visual size assessments were done to allow rapid recordings of animals that were manually removed from the net and directly discarded by the fishermen. A total of 10 cm size bins were chosen to facilitate length class allocation within a logistically difficult sampling environment and to ensure consistencies between observers. Species were classified as belonging to either the commercialized or non-commercialized fractions. It needs to be noted that during the retrieval of the trammel nets part of the organisms that have been entangled in the net may become dislodged during the process falling back into the sea before reaching the boat. As the dislodgment of entangled individuals can happen below as well as above the water this fraction cannot be quantified during normal commercial activities. Quantifying this part of the catch would require dedicated complex sampling involving scuba divers and was outside the scope of the current study. The abundances and biomass of catches presented are therefore relative values of the quantities caught by the fishing gear used. These may underestimate the true absolute quantities of organisms that did get entangled (caught) by the net. This data limitation is an inherent attribute of commercial catch and bycatch studies. In addition, the survival status (i.e., alive or dead) was noted for each discarded animal as the gear was hauled on board; where "dead" animals were observed to have no reflexes or active movement following the protocol by *Benoit, Hurlbut & Chasse (2010)*, described below in more detail.

During net retrieval, small invertebrates or bed-forming organisms (e.g., *Posidonia oceanica*) were passively disentangling and falling on the deck in a continuous pattern; therefore, it was not possible to assign this fraction of the catch to a specific netting wall. Instead, this fraction was recorded at the level of the whole net and quantified by determining its total volume in liters. A subsample of 20 L in volume was taken and kept for further analysis at the laboratory from which all items were identified to species level when possible and weighed. All data were organized into a hierarchical (fishing trip, net, netting wall, and item) database.

## Representativeness of the sampled boats

Due to logistical constrains, only three vessels were sampled. Therefore, to ensure the representativeness of this sample, the commercialized catches of the three surveyed boats were compared to catches made by the remaining fleet targeting lobster. Weight (kg) and first sale price (Euros/kg) for any sale of the entire Majorcan fleet are easily available because all sales of fish are carried out in the single fishing wharf of the island. The
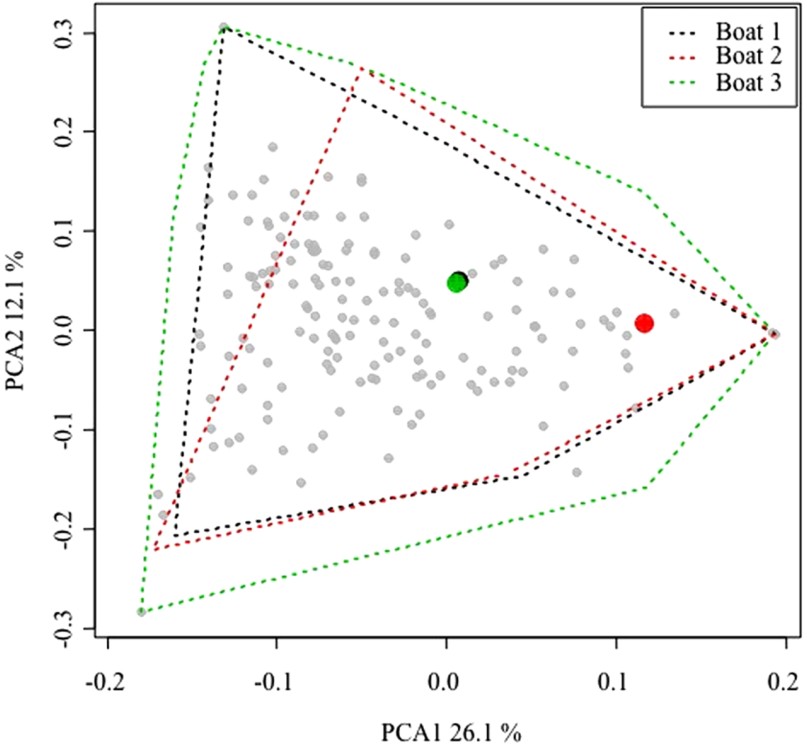

**Figure 2 Principal components analysis (PCA) plot of all the small-scale boats from Majorca that targeted spiny lobster during the 2015 and 2016 spiny lobster seasons.** Between-boat similarity was estimated from the landed catches (kilogram of each one of 170 commercial categories) for all the daily fishing trips of the lobster trammel net fleet from Majorca. The gray points denote the average scores for all the boats (i.e., the centroid of all the fishing trips of a given boat). The larger, colored points denote the three sampled boats. For them, an envelope polygon including all the fishing trips of each boat was added. Note that within-boat variability is similar to or even larger than between-boat variability. The number on the axis label indicates the proportion of variance explained by each of the first two PCA axes considered.                                 

landings data can be disaggregated by commercial category, day and boat and thus the commercialized catch is known for every fishing trip (*Palmer et al., 2017*). Fishing trips with trammel nets targeting lobster were filtered out using the algorithm described by *Palmer et al. (2017)*. The data corresponding to 2015 and 2016 (the same period as this study sampling) were used in a principal component analysis (PCA) using the *rda* function of the *vegan* library (*Oksanen et al., 2015*) in the R package (*R Core Team, 2008*) on Hellinger-transformed data (*Borcard, Gillet & Legendre, 2011*; *decostand* function of vegan library). The similarity patterns in landings at the between-boat and within-boat level were depicted and assessed using a plot of the scores in the first two dimensions of the PCA space.

After compiling all the fishing trips of the lobster trammel net fleet, the within-boat variability in terms of landings composition (i.e., landed biomass of 170 commercial categories; *Palmer et al., 2017*) was similar between the sampled boats and the rest of the fleet (Fig. 2). The convex hull polygons that include all the fishing trips of the three sampled boats covered the same spatial area in the graph as the average scores of the

remaining boats in the fleet. The average scores of the three commercial boats sampled were found well within the space occupied by the average scores of the remaining fleet. This indicates that within-boat variability with respect to the season is comparable or even greater than between-boat variability. Therefore, the three sampled boats could be considered representative of the small-scale Majorcan lobster fishing fleet.

## Revenue patterns at the netting wall level: effects of manufacturing material and the use of *greca* on commercial catches

Revenues from the commercialized fraction (i.e., gross revenue corresponding to the first sale in Euros) for any single individual were estimated from the weight and average price of the corresponding commercial category. Sale prices of a given category were estimated after averaging the sales prices of this category over all the trammel net fishing trips corresponding to the 2015 and 2016 spiny lobster fishing seasons.

After preliminary inspection of the cumulative (across species) revenue per netting wall, the observed distribution seemed to follow the same pattern as the biomass amount. Therefore, the same statistical model and Bayesian fitting approach described in the next section were adopted.

## Patterns of discards at the trammel and netting wall levels: effects of manufacturing material and the use of *greca*

Between-species comparability in terms of abundance was problematic because discards ranged from large fish (e.g., rays) to fragments of reef-forming algae that cannot be enumerated as individual organisms. Due to the logistical constraint of sampling on board of commercial vessels weight measurements could not be attained for fish thus total length measurements to the nearest centimeter were preferred for most individuals. To circumvent this problem, fish catches were converted to biomass (kg). Published length/weight relationships were used for this conversion (*Morey et al., 2003*; *Ilkyaz et al., 2008*; *Bilge et al., 2014*). However, when precise length measurements were not practical the organisms were assigned into 10 cm length classes (see justification above), except for *P. elephas* which were assigned into 5 cm CL classes. For the calculation the maximum of each length class category was used. In the absence of a size record for an individual item within a catch, the average size of the sampled conspecifics was used to estimate its weight. Due to the low precision of visual length estimates and the absence of size data for occasional individuals as aforementioned the calculated biomass values need to be viewed with this limitation in mind. For the invertebrate bycatch fraction weight was determined via a subsample in the laboratory on land (see details above).

After preliminary inspection of the cumulative biomass (i.e., pooling weight across species) per netting wall, the distribution of the data for netting walls in which some discard was reported seemed to follow a negative exponential distribution, but the

number of netting walls without discards largely exceeded the expected value. Therefore, the following model was considered:

$$x_i \, \mathrm{dexp}\left(\frac{1}{\mathrm{mean}_i}\right)$$

$$\mathrm{mean}_i = (\mathrm{mean.wall}_{\mathrm{type}} + \mathrm{mean.rnd}_{\mathrm{boat}} + \mathrm{mean.rnd}_{\mathrm{net}})w_i$$

$$w_i \, \mathrm{dbinom}\left(p_{\mathrm{type}}\right)$$

where $x_i$ is the discarded biomass of the netting wall $i$; $w_i$ is an indicator variable (zero in case of no catch, one otherwise); $\mathrm{mean}_i$ is the inverse of the characteristic parameter of the exponential distribution for netting wall $i$, which results from combining mean.wall$_{\mathrm{type}}$, a fixed effect related to the type of manufacturing material (or the use of *greca*); and mean.rnd$_{\mathrm{boat}}$ and mean.rnd$_{\mathrm{net}}$ are random effects that accounted for the fact that all netting walls from a given boat or net were structurally linked. The strings dexp and dbinom indicate the assumed underlying distributions for the exponential and binomial distributions, respectively. The parameters mean.rnd$_{\mathrm{boat}}$ and mean.rnd$_{\mathrm{net}}$ were assumed to be normally distributed with zero mean across boats or nets. Two separate sets of analyses were completed: (i) PFM vs. MMF and (ii) MMF vs. MMF + *greca*.

The parameters of the model were fitted using a Bayesian approach. Samples from the joint posterior distribution of parameters given the data were obtained using JAGS (*Plummer, 2003*; http://mcmc-jags.sourceforge.net/). Virtually non-informative priors were set. A custom script calling JAGS via *R2jags* library from the R package (*R Core Team, 2008*) was used. Three Markov chain Monte Carlo were run and 30,000 posterior samples were retained after proper burning (10,000) and thinning (1/10). Convergence was assessed by visual inspection of the chains and tested using the Gelman–Rubin statistic (*Plummer et al., 2006*). Thereafter, according to a Bayesian framework, effects attributable to a given treatment were labelled *statistically not relevant* or *statistically relevant* (corresponding to "statistically non-significant"/"statistically significant" of classic inferential statistics) depending on whether their corresponding 95% Bayesian credibility intervals overlapped or not with a targeted reference value.

In addition to the univariate analyses described above, differences in species composition attributable to the manufacturing material or *greca* use were tested using a redundancy analysis (RDA; *Borcard, Gillet & Legendre, 2011*). The analyses were completed using the *rda* function of the *vegan* library (*Oksanen et al., 2015*) in the R package on the Hellinger-transformed data (*Borcard, Gillet & Legendre, 2011*; *decostand* function of the vegan library). Between-category differences were assessed using the permutation procedure implemented in the *anova* function from the *vegan* library; thus, net level random effects were ignored.

Note that all the analyses above refer to the netting wall level, but some discards, i.e., invertebrate fraction could not be assigned to a given netting wall but only at the entire trammel net level (see above). Thus, both fractions (i.e., items attributable to netting wall level and those attributable to the net level) were pooled at the net level for a second set of analyses. After preliminary inspection of the distribution of

**Table 1 The number of animals discarded that were alive at onboard arrival.**

| Species | Number of observed animals | | Immediate survival probability | Bayesian 95% credibility interval | |
|---|---|---|---|---|---|
| | Alive | Total | | 2.5% | 97.5% |
| *Palinurus elephas* | 82 | 127 | 0.64 | 0.56 | 0.72 |
| *Parastichopus regalis* | 30 | 33 | 0.92 | 0.80 | 0.98 |
| *Leucoraja naevus* | 193 | 296 | 0.65 | 0.60 | 0.71 |
| *Raja* sp. | 5 | 229 | 0.02 | 0.01 | 0.04 |
| *Raja clavata* | 26 | 224 | 0.11 | 0.08 | 0.16 |
| *Scyliorhinus canicula* | 12 | 161 | 0.07 | 0.04 | 0.12 |
| *Scorpaena scrofa* | 4 | 100 | 0.04 | 0.01 | 0.08 |
| *Lophius piscatorius* | 1 | 46 | 0.02 | 0.00 | 0.08 |

**Note:**
Only species with more than 30 observations are detailed. The immediate survival probability and 95% Bayesian credibility intervals are indicated.

log-transformed cumulative (across species) biomass (kg) per trammel net, the data appeared to be normally distributed, which allowed a simpler analysis in comparison with the analysis at the netting wall level. After removing all mixed nets (i.e., those with netting walls composed of more than one fiber type), the differences attributable to fiber type in the log-transformed biomass were estimated using the function *lm* in R. Note that only MMF vs. PMF was compared because the *greca* was always set in mixed nets.

In the analytical setup described above, using netting wall as a statistical unit, it can be deduced that consecutive netting walls cannot be considered as statistically independent. To address this concern the data were tested for the existence of spatial (i.e., between consecutive netting walls) autocorrelation which was rejected justifying the use of the presented analyses at netting wall level (For the test and its results see Fig. S1).

## Survival assessment

Survival was assessed at two temporal scales: immediate and short-term (seven days). The aim of the first method was to determine the immediate survival of the animals to be discarded when the gear arrived on deck. The time that each animal had been retained in the net was unknown but was less than the maximum soaking time (48 h). The retrieval of the trammel net is relatively quick, and the air exposure of the animals to be discarded during processing is usually less than 2 min. Animals to be discarded were identified and measured, and their survival status was assessed (see survival assessment method below). The following species were caught in sufficient numbers (>30 individuals) to permit a formal analysis: spiny lobster (*P. elephas*), cuckoo ray (*Leucoraja naevus*), unidentified species of skates (*Raja* sp.), thornback ray (*Raja clavata*), lesser spotted dogfish (*Scyliorhinus canicula*), red scorpionfish (*Scorpaena scrofa*), angler fish (*Lophius piscatorius*) and royale cucumber (*Parastichopus regalis*). The proportion of living animals at deck arrival was estimated from the ratio of living to total observed animals in the unwanted catch, and the Bayesian 95% credibility interval around that proportion was calculated using an ad hoc R script (Table 1).

Additionally, short-term survival (seven days) was assessed for the target species only. A total of 16 discarded specimens of undersized lobsters collected in Port d'Andratx between 26/5/16 and 16/8/16 were transferred into small, aerated and refrigerated tanks and transported to an onshore laboratory (LIMIA facilities). The collection of juvenile lobster specimens for survival test was carried out under specific permission from the Balearic Government. In the laboratory lobsters were placed in 5,000 L fiberglass tanks and supplied with a continuous flow of 100 μm filtered and refrigerated water; the temperature was maintained below 18 °C. The tank was also equipped with plastic tub shelters. Tanks had a maximum occupancy of five individuals to avoid stress and aggressive behavior among individuals. The individuals were fed once per day with fish. The vitality status of the specimens was determined using a four-point vitality scale (*Benoit, Hurlbut & Chasse, 2010*; Table S1) which is based on an evaluation of damage and reflex impairment closely related to the reflex action mortality predictor proposed by *Davis & Ottmar (2006)* on days zero, one, two, four and seven. At the end of the seven days observation period, all surviving individuals were then marked with "T" marks and returned to the sea in a marine protected area. The small lobsters were submerged by scuba divers at a depth of 40 m and released at natural shelters to avoid predation.

A Kaplan–Meier survival function was calculated for the spiny lobster by combining immediate and short-term survival data. For estimating survival probability and 95% confidence intervals, the surviving animals were assumed to be censored at day zero (i.e., when they came on board). The analysis was conducted using the *survival* package (*Therneau & Grambsch, 2000*) in R.

## RESULTS

### General catch patterns

A total of 1,550 netting walls corresponding to 70 trammel nets and 35 fishing trips were surveyed. The fishing depth ranged from 63 to 130 m (mean depth ± S.D.: 94.4 ± 19.3 m). The average soaking time was 45 h. (2.3 ± S.D). The handling of the catch was generally fast, and the disentangling of items from the net took 2 min maximum.

A total of 706 animals, from 19 species, comprised the wanted/marketable catch (Fig. 3A). As expected, the most frequently recorded species was lobster, but other highly marketable species such as *S. scrofa* (17.68 Euros/kg) or *Zeus faber* (20.94 Euros/kg) were also frequently caught. *R. clavata* was the third most common marketable species.

The average number of commercialized items per netting wall was low. The 95% quantiles of the number of animals per netting wall ranged from 0 to 3 with a median value of 0. The biomass (kg) ranged from 0 to 4.4 kg with a median of 0 and a mean of 0.61 kg per netting wall. Finally, the 95% quantiles for revenues (Euros) ranged from 0.0 to 103.92 Euros with median of 0.0 and a mean value of 12.75 Euros per netting wall.

The total number of discarded species was 47. The number of discarded animals recorded at the netting wall level was 1,465. The most frequent discards were elasmobranchs (Fig. 3A). Specifically, the three most frequently discarded species were rays (Rajidae). It is also noteworthy that the 5th most commonly discarded species was the target lobster, where the discarded specimens were undersized. This species also

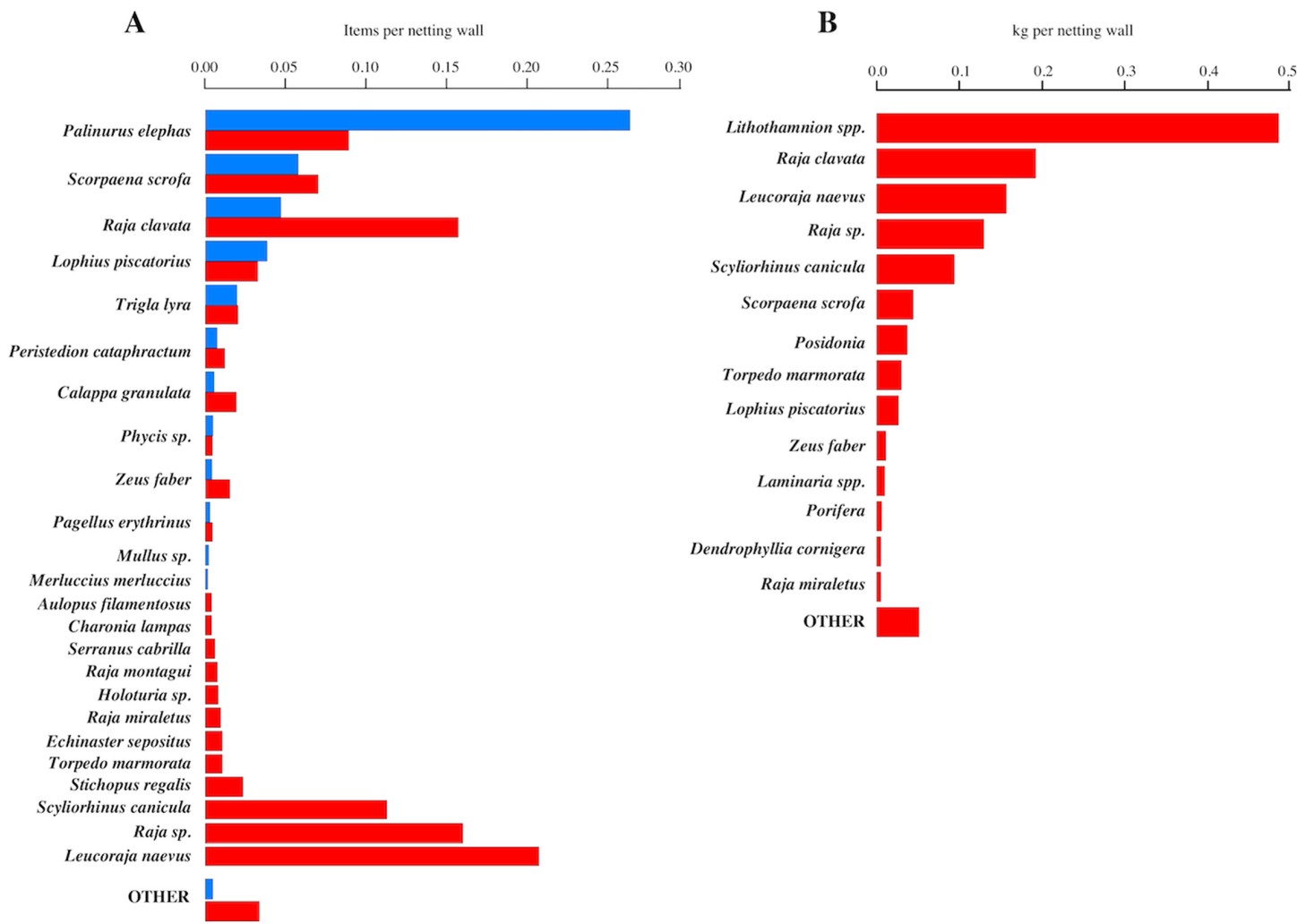

**Figure 3  Marketable catches and Discards.** (A) Number of items per netting wall of Marketable catches (blue bars); and Discards non-marketable species (red bars); (B) kilogram per netting wall of discards (non-marketable species), pooled for catch attributable at the netting wall/panel and whole net levels.                                                            

was the main discard species with 6.35% following the EU landings obligation criteria. The "other" group included 24 different species, each one with less than five items per species. The number of discarded items per netting wall ranged from 0 to 4 (95% quantiles), with a median of 1. The biomass (kg) per netting wall ranged from 0 to 4.6 kg, with a median of 0.03 kg and a mean of 0.68 kg per netting wall.

The assemblage of non-marketable/discarded fraction at the net level (i.e., pooling animals at the netting wall level with the fraction falling on deck) differed from that seen at the net panel level (Fig. 3B). At the pooled net level, the most common species were the reef-forming, calcareous algae of the genus *Lithothamnion* ssp. (maerl) followed by four elasmobranchs. The overall discard faction by weight amounted to 69.74%. Note, that the units in Fig. 3B are kg and not items in order to facilitate between species comparison, i.e., inclusion of invertebrate and plant fraction. Thus *Lithothamnion* spp.

have a proportionally larger weight compared to other organisms causing a bias in a direct comparison. Therefore, excluding maerl the discard faction by weight was 52.71%. Besides calcareous algae the discards also contained other habitat-forming organisms such as Seagrass (*P. oceanica*), Ascidians (*Botryllus schlosseri*), Kelp (*Laminaria* spp.), Sponges (Porifera) and hard corals (*Dendrophyllia cornigera*), indicating that the fishing gear interacted with the benthos.

## Effects of manufacturing material and the use of *greca* on marketable catches

Regarding the MMF vs. PMF comparison (Fig. 4A), the proportions of netting walls/panels with some marketable catch were similar: PMF = 0.31 (95% Bayesian credibility, interval between 0.28 and 0.34) vs. MMF = 0.28 (95% Bayesian credibility, interval between 0.24 and 0.32). The estimated mean revenue when marketable items were reported in the netting wall was also similar: PMF = 41.5 Euros (32.9–50.3) vs. MMF = 42.3 Euros (31.5–52.6). The mean revenue for an average net (22 netting walls), after combining the two different estimations to calculate the expected mean commercialized catch per net, was PMF = 262.1 Euros/net (63.5–504.2) and MMF = 242.5 Euros/net (44.9–563.8). In all three cases, the differences between PMF and MMF were not considered statistically relevant.

Concerning the standard MMF vs. MMF + *greca* comparison (Fig. 4B), the differences in the probability of obtaining some commercial catch were statistically not relevant: MMF = 0.33 (0.23–0.44) vs. MMF + *greca* = 0.44 (0.33–0.56). Similarly, the estimated mean revenue when marketable items were reported in the netting wall did not to differ: MMF = 40.5 Euros (27.5–62.5) vs. MMF + *greca* = 39.5 Euros (27.9–57.7). After combining these estimates, the differences in the expected (mean) revenues from the pooled net (22 netting walls) was −90.0 Euros/net favored the MMF + *greca*, but this difference was not statistically relevant (95% quantiles ranging between −540.9 and 331.7 Euros/net).

## Effects of manufacturing material and the use of *greca* on non-marketable/discarded catch

Concerning the MMF vs. PMF comparison at the netting panel level (Fig. 4C), the proportions of panels with unwanted catch were similar: PMF = 0.50 (0.47 and 0.51) vs. MMF = 0.49 (0.44 and 0.53). The estimated mean weight of discarded animals was also similar: PMF = 1.38 kg (0.70–1.47) vs. MMF = 1.34 kg (1.02–1.71). The differences in these two variables attributable to net type were not relevant. The expected (mean) weight of discards at the net level (i.e., 22 netting walls pooled; the average number of netting walls per net) after combining the two model parameters were PMF = 14.0 kg/net (5.7–27.6) vs. MMF = 13.5 kg/net (4.9–26.9). These differences were not statistically relevant.

Comparing the standard MMF vs. MMF + *greca* at the netting wall level showed relevant differences in the probability of netting walls with unwanted catches (Fig. 4D): MMF = 0.52 (0.41–0.63) vs. MMF + *greca* = 0.70 (0.58–0.79). The estimated mean

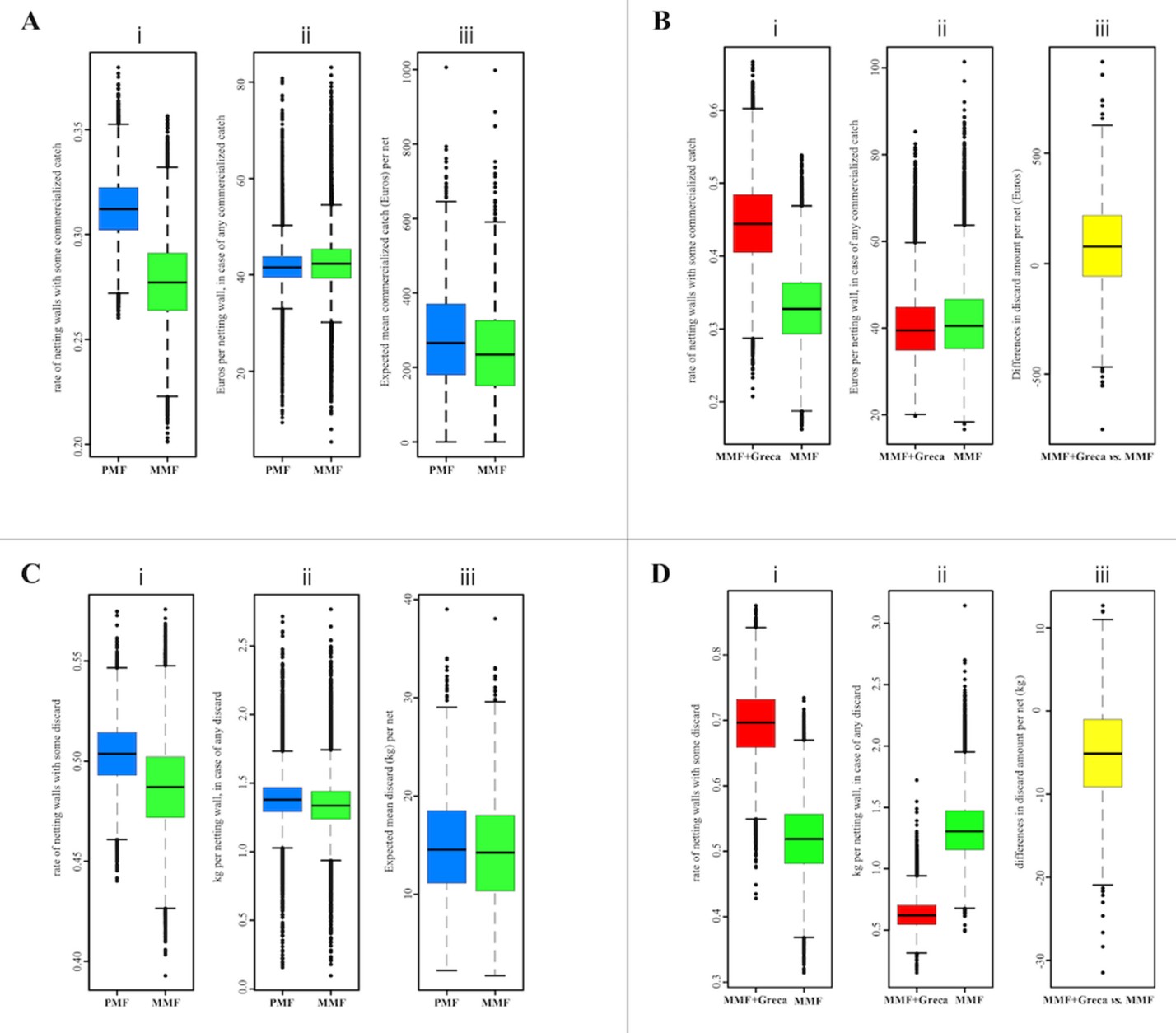

**Figure 4 PMF vs. MMF (A and C) and MMF vs. MMF + *greca* (B and D) comparisons.** Comparing PMF vs. MMF (A and C) and MMF vs. MMF + *greca* (B and D) by (i) the ratio of netting walls with some catch; (ii) the revenue (in Euros) of the commercialized catches (A and B) or the weight of the discards (C and D) per netting wall with some catch; and (iii) the revenue (in Euros) of the commercialized catches (A and B) or the weight of the discards (C and D) for an average (22 netting walls) net. Only items attributable to a given netting wall were used in these comparisons.

discarded weight was statistically relevant, but in the opposite direction: MMF = 1.30 kg (0.90–1.86) vs. MMF + *greca* = 0.62 kg (0.43–0.91). Thus, MMF + *greca* netting walls tended to retain some discarded fauna more frequently, but the overall mean weight of discards was smaller. At the net level, the estimated amount of discard per net was smaller

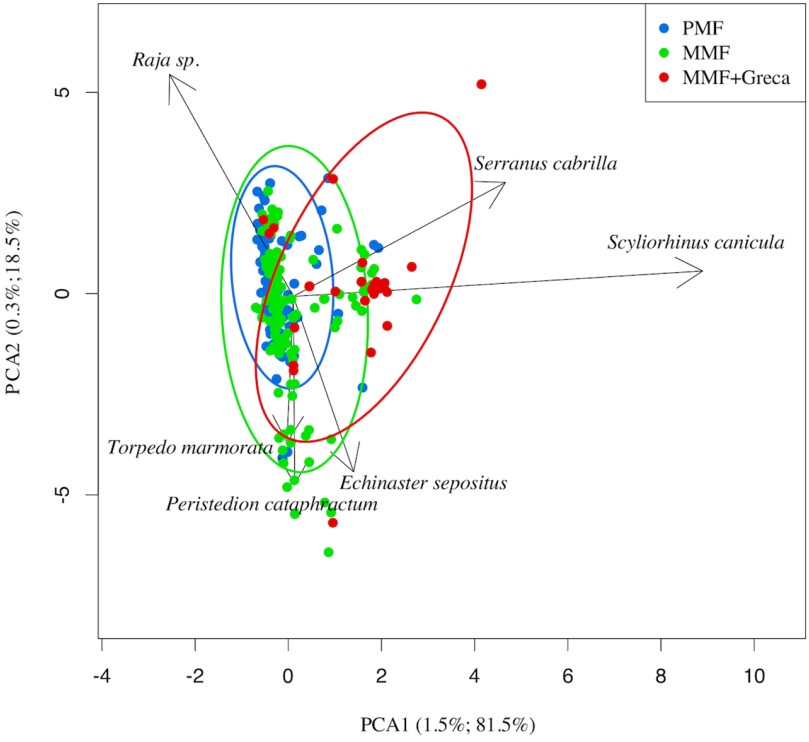

**Figure 5 PCA of the discarded items attributable to a given netting wall (points in the figure).** The PMF and MMF largely overlap, but MMF + *greca* significantly differs in species-specific abundance. This difference is mainly related to a higher abundance of *S. canicula*.

for MMF + *greca*, but the difference was not statistically relevant (prob. = 0.20; mean difference: 5.43 kg; 95% CI between 18.73 and −5.6 kg).

The RDA results for the differences in species composition are shown in Fig. 5. Differences between net types were statistically significant (variance explained: 0.50, residual variance = 21.49; $F = 9.05$ d$f$ = 2,767; probability after 1,000 random permutations < 0.001). The main differences in species composition between MMF/PMF vs. MMF + *greca* was due to the relative abundance of *S. canicula* (which has a smaller body size and was more frequently caught with MMF + *greca*) (Fig. 6A) and Rajidae species (which have a larger body size and were more frequently caught with MMF/PMF) (Fig. 6B).

The second set of analyses including all discards were pooled at the trammel net level, standardized by the number of netting walls. In this case, the sample size considered (i.e., number of nets) was smaller (44 nets) because nets with more than one type of manufacturing material were removed, which is the case for all MMF + *greca* netting walls. The estimated mean biomass of discards for PMF was 1.19 kg/netting wall (95% CI [0.94–1.51]). The corresponding figures for MMF were mean = 0.92 kg/netting wall (0.63 and 1.35). In that case, the differences between MMF and PMF in discards weight (kg) per trammel net, after accounting for differences in the number of netting

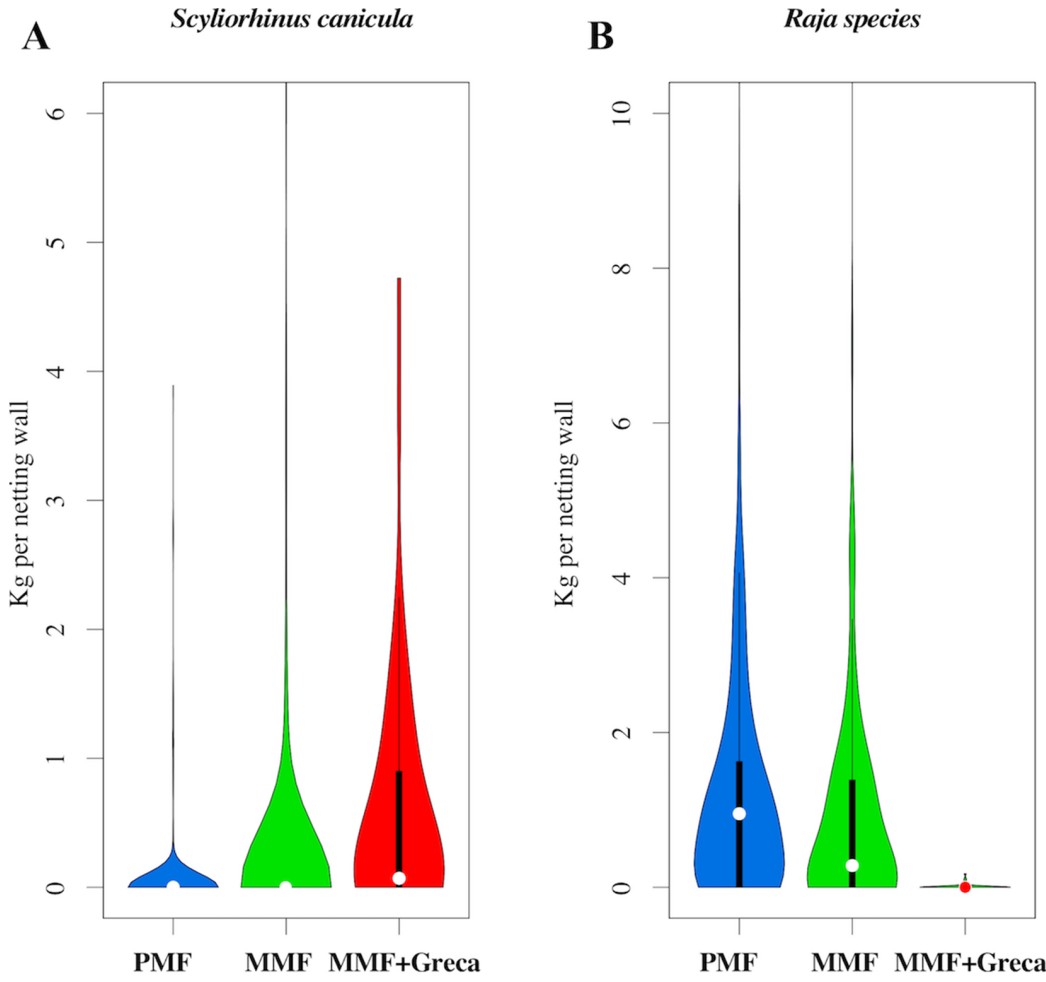

**Figure 6 Abundance distribution (kilogram per netting wall) of the species that contributed most to the differences in the discarded fraction between the different types of manufactured netting walls.** (A) *S. canicula*; (B) Raja species.

walls per net, were not significant (effect of the net type: $df = 1$, SS = 0.57; residual: $df = 41$, SS = 17.67; $F = 1.34$, $p = 0.25$) (Fig. S2).

## Immediate and short-term survival assessment

A total of 1,216 animals from eight species were examined at arrival on board, and 353 were alive (Table 1). The immediate survival probabilities of the most abundant discarded species ($n > 30$; *P. elephas*, *L. naevus* and *P. regalis*) were all greater than 0.6. In contrast, the survival probability of other commonly discarded species was less than 0.2 (Fig. 7).

A sample ($n = 16$) of living, undersized spiny lobsters (mean CL = 7.1 ± 0.9 cm) was kept in captivity for seven days. From that sample, only one lobster died that already had a "poor" vitality status on day zero. In contrast, the animals that were classified with a "good" vitality status on day zero quickly improved vitality status to "excellent" (Fig. 8).

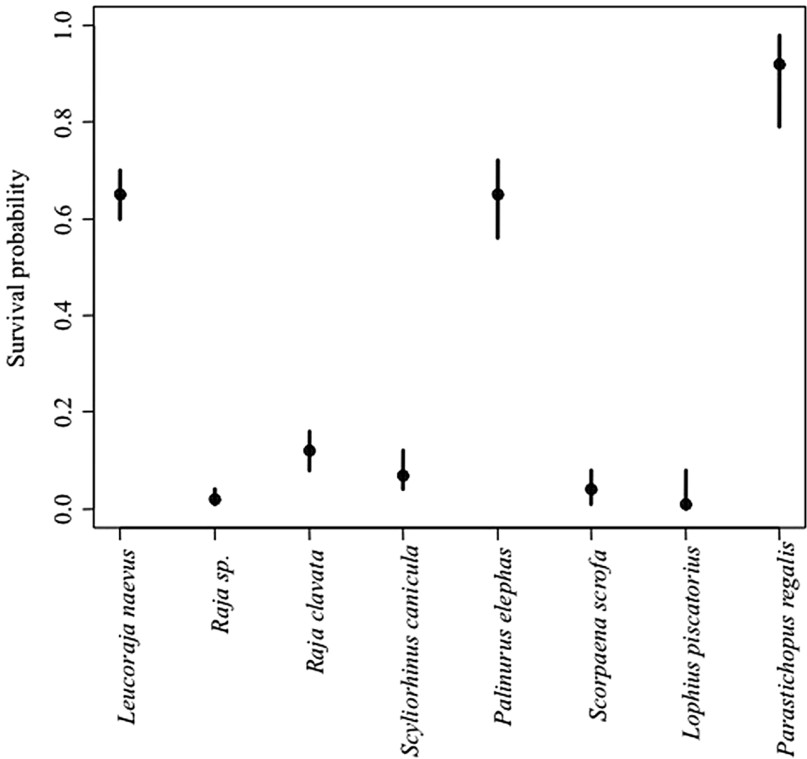

**Figure 7 The proportion of discarded animals that were alive when they arrived on board.** The Bayesian 95% credibility intervals are indicated.

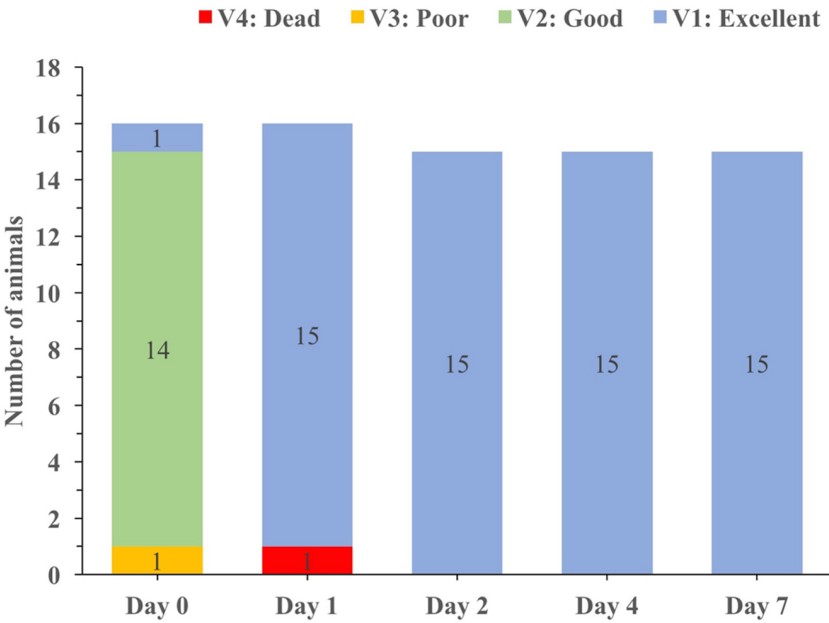

**Figure 8 Changes in vitality status over time for the spiny lobsters caught and held in captivity for up to seven days.**

The data from immediate and short-term survival was combined using the Kaplan–Meier survival function (Fig. S3). The survival curve reached an apparent asymptotic value after one day in captivity, which provided an asymptotic estimate of survival probability (0.64; 95% CI between 0.54 and 0.76).

## DISCUSSION

Within the present study we compared three trammel net designs (PMF, MMF and MMF + *greca*) in terms of biomass, species composition and revenue of marketable catches and discards. The statistical analyses suggested no differences between PMF and MMF in terms of probability of obtaining commercial catch or in cumulative (across-species) biomass/catch weights. The results for the non-marketable/unwanted fraction were statistically indistinguishable. Similarly, the comparison between MMF + *greca* and MMF showed no differences for the wanted and unwanted fractions. In summary, the new designs do not present an economic disadvantage nor a relevant reduction of the cumulative discards across species. These results disagree with those reported for the Mersin Bay small-scale fishery (*Gökçe et al., 2016*). In that case, using a *greca*-like guarding net on the lead line of prawn trammel nets reduced the capture of discard species and the time required to clean the nets (*Gökçe et al., 2016*). Therefore, the usefulness of a given technological novelty is probably case-specific.

The only reputed advantage of these nets, which has not been evaluated here, is the potential reduction of expenses related with repair and maintenance. In this sense, damage reduction has been reported when *greca* was used in red mullet (*Mullus* spp.) and shrimp (*Melicertus kerathurus*) fisheries at the Aegean Sea (*Metin et al., 2009*; *Aydin, Gökçe & Metin, 2013*). In these cases, the reduction of net damage was attributed to a reduced bottom contact of the net with the seabed and the avoidance of capturing benthic invertebrates that appear to be selectively excluded by the *greca*. Benthic invertebrates such as starfish, crabs or emergent fauna can damage trammel nets by adding undesired weight to the bottom part, thus increasing seabed contact and wear of the net. Furthermore, undesirable invertebrates once caught need to be disentangled increasing the likelihood of wear and damage to the net during this process.

The results reported here for the *greca* indicate that it affects the capture of some non-commercialized species. It reduces capture of rays but increases the capture of other smaller elasmobranch species within our study. The processes leading to this pattern remain elusive, but the phenomenon illustrates the need to more closely examine the operational mechanism of the trammel net. In this type of net, the netting wall is formed by a main net characterized by a smaller mesh size and a larger diameter thread at the center with two further nets on either side, both with wider meshes sizes and smaller thread diameter. When a fish tries to pass through this netting wall it pulls at the central panel, which is slack, and when it tries to escape through the wider lateral meshes, it becomes trapped. Because of the extra weight that the *greca* exerts on the netting wall, the meshes may acquire rhombic forms, and reduce the level of slack in the panel effecting the hanging ratio of the nets. Such circumstances could explain the differences in species composition mentioned above. Bigger and flatter fishes such as *Raja* spp.

would have less chance to be trapped due to the tension of the panel and to the deformation of the meshes. In contrast, smaller elasmobranchs such as *S. canicula*, due to their fusiform morphology, would penetrate the exterior panels and become trapped more easily in the more tensioned netting walls fitted with *greca*.

The targeted spiny lobster species is generally caught by the trammel net when they are attracted by the carcasses of previously trapped fish. This mechanism requires that the soaking time of lobster trammel nets tends to be longer than for any other type of trammel net. This long soaking time determines three relevant aspects in catch: (i) the amount and composition of bycatch and discards, (ii) the survival potential of caught species and (iii) catch revenues. All three aspects are discussed in the next paragraphs.

The first aspect is that the non-marketable fraction of catch from lobster trammel nets tends to be larger than in other small-scale fisheries in Mediterranean and Black Sea (Area FAO 37) which have been reported to discard between 5% and 43% of the catch (*Tsagarakis, Palialexis & Vassilopoulou, 2014*). If we consider only discards as species defined by the EU landings obligation, our results indicate a discard ratio of 6.35% (with discards being mainly comprised of undersized lobsters). This contrasts with 52.71% of discarded biomass if all organisms except maerl (Coralline algae) were included and 69.74% if maerl was included. Similar high discard ratios of over 40% for trammel net fisheries were reported for lobster fisheries by *Quetglas et al. (2004)* and for prawn fisheries by *Gökçe & Metin (2007)*. Trammel net fisheries targeting mainly fish appear to have lower discard ratios. *Stergiou, Moutopoulos & Erzini (2002)* for example reported 5.1% discards and *Gonçalves et al. (2007)* 14%. While the values reported by the present study appear significantly higher compared to other studies it needs to be considered that not all studies have included the invertebrate and plant fraction in their discard calculations. Furthermore, various studies report the percentage of discarded items which will be distinct from the biomass values reported herein. Since no standard reporting protocol exists not all reported discard percentages are strictly comparable.

Considering the % discard levels reported for this study (see above) and the presence of habitat-forming organisms (e.g., maërl), the lobster trammel nets are clearly interacting with benthic communities. While the experimental design implemented within this study does not allow conclusions on the long-term effects of trammel nets on the benthos dynamics, they do provide some indication of possible impacts. For example, maërl (calcareous red algae), which represented the largest component by weight, while not negatively affected by fragmentation, i.e., maërl propagation is facilitated by fragmentation (*Irvine & Chamberlain, 1994*) it is not tolerant to desiccation due to their low water content and lack of protective mucilage. As a consequence, maërl show a significant decrease in survival when the thalli are out of the water for longer than 5 min (*Wilson et al., 2004*). This is a very likely time frame for unwanted catches to remain on deck considering current fishing practices. This potential effect on maërl may be mitigated by adapting fishing techniques to reduce capture and by returning maërl fragments caught in the net immediately to the sea. While impacts of trammel nets on benthic habitats are thought to be considerably lower than impacts from trawls fisheries, pots or traps are considered the least damaging activity (*Grieve, Brady & Polet, 2014*).

Traditionally traps were the main fishing method adopted in the Balearic Islands. However, the catch rates for traps in the Balearic Islands are lower than for trammel nets (*Amengual-Ramis et al., 2016*) and thus traps have been discarded by the fishing sector as being less economical and operational viable. Ultimately, the magnitude of damage by trammel netting will depend on the intensity of fishing operations within an area and its benthic community composition (*Hinz, 2016*). More research is needed to quantify the actual disturbance caused by trammel nets on benthic habitats in the Balearic Island and elsewhere. To limit the effect of trammel netting on habitats operational mitigation measures should be investigated in conjunction with the fishing sector.

The second aspect is related to the survival probability of the non-marketable fraction. Some animals in the catch are alive when they arrive on board and may survive if returned to the sea. The reasons for deciding that a given animal is not worth landing and can be returned to the sea are diverse. For example, adherence to the local management rules demands that undersized lobsters cannot be retained. The fraction of undersized lobsters (CL <90 mm) can be up to 80% in Sardinia (Italy) (*Secci et al., 1999*); however, in the current study this fraction was 25.5%, and a similar figure (21.5%) was reported by *Quetglas et al. (2004)*. Considering the capture of undersized lobsters one of the more relevant results reported here, although based on a relative small number of samples, was the high survival probability of undersized *P. elephas* estimated at 0.64 after combining immediate and short-term (seven days) survival in captivity. This result may be indicative of the potential survival of undersized individual if returned to the sea. Further, other non-marketable species were reported to have high survival (>60%) when they were brought on deck: cuckoo ray (*L. naevus*) and the royale cucumber (*P. regalis*). *P. elephas* and *P. regalis* are both invertebrates, which may make them more resilient to asphyxiation and attacks from predators and scavengers. However, it is interesting that an elasmobranch species, *L. naevus*, also demonstrated a high survival. In contrast, other non-commercial species showed low survival capability. This is likely to be due to the prolonged soaking times of this fishery (up to 48 h) that can result in exhaustion, asphyxiation and injury of animals caught in the net, as well as by predation from attracted predators and scavengers. As mentioned above, these prolonged soaking times are a deliberate part of the fishing practice. Therefore, alternative gear configurations and/or fishing tactics that reduce soaking time would be preferable and should be investigated. Furthermore, based on the results from the medium-term survival experiments of spiny lobsters, it is recommended that further research should be conducted that may support a potential exemption from the LO of this species with respect to article 15, paragraph 4b, of the CFP (EU Regulation, 1380/2013, *European Union (EU)*). Returning ovigerous females and undersized lobsters to the sea could provide substantial benefits to this exploited stock.

The third aspect related to the lobster trammel net fishing strategy concerns revenue. The economic relevance of the trammel net fishery from Majorca not only relies on the target species but also on several species of intermediate or high economic value, and this confirms that this and other small-scale fisheries in the Mediterranean target multiple species. In this case, only 18% of the catch corresponds to the target species

and 14% to other marketable species. Therefore, fishers appear to modulate their fishing tactics in response to the pooled revenue of their catch. Aiming to explain fishers behavior only considering the nominal target species may thus be naïve. For example, a long net soaking time may increase lobster catches, but the marketable condition of both lobster and other catch species is likely to improve by reducing the soaking time (<48 h). Another trade-off may emerge between expected revenues and extra work that may change the decisions of fishers on whether to land low-value species. In this case, the only species that was frequently captured and occasionally marketed was thornback ray (*R. clavata*). For this species, the probability of discarding an individual significantly depends on size (the larger the fish, the smaller the discarding probability; the size at which the discarding probability is 50% is 53.4 cm) and on the number captured during a given fishing trip (more fish captured means smaller discard probability, and the number of fish at which the discarding probability is 50% is 26 cm). Apparently, from the fisher's perspective, the extra work of sorting a new marketable species is only worth the effort for large catches, in terms of the size or number of fish caught.

## CONCLUSION

In summary, the strength of the interactions with benthos may be greater for the trammel net fishery than for other small-scale métiers. Possible management measures for reducing interactions may be developing new fishing technologies that reduce soaking time while, at the same time, maintain revenues. However, any management measures should be adopted in conjunction with the fishing sector. Despite the fact that most small-scale boats in Majorca alternate different fishing strategies (or métiers) throughout the year, the lobster season represents a substantial percentage of the annual revenues (*Palmer et al., 2017*). Therefore, new management rules affecting the lobster trammel net fisheries may affect the whole small-scale fleet, because most boats in Majorca may be at the limit of their economic viability. This statement is supported by the negative trend in boat numbers experienced in recent decades (*Grau et al., 2015*). The fact that landings remain constant while revenues are continuously decreasing suggests that the dynamics of the fleet are not driven by the resource dynamics but by a marketing strategy that may be improved (*Palmer et al., 2017*). Therefore, new management plans should only be enforced after careful evaluation of both empirically demonstrated effects on the long-term dynamics of benthic habitats and on the economic viability of the small-scale fleet.

## ACKNOWLEDGEMENTS

This study is a result of the Associated Unit LIMIA-IMEDEA. The authors thank the fishermen of Port d'Andratx, Portopetro and OPMallorcaMar for their collaboration in this study.

### Funding

This research received funding from the European Commission's Horizon 2020 Research and Innovation Programme under Grant Agreement No. 634495 for the project Science,

Technology, and Society Initiative to minimize Unwanted Catches in European Fisheries (MINOUW). Gaetano Catanese was supported by a research contract from DOC INIA-CCAA program. Hilmar Hinz was supported by the Ramón y Cajal Fellowship (grant by the Ministerio de Economía y Competitividad de España and the Conselleria d'Educació, Cultura i Universitats Comunidad Autónoma de las Islas Baleares). Andrea Campos-Candela was supported by a FPU pre-doctoral fellowship (ref. FPU13/01440) from the Spanish Ministry of Education, Culture and Sports (MECD). The funders had no role in study design, data collection and analysis, decision to publish, or preparation of the manuscript.

### Grant Disclosures

The following grant information was disclosed by the authors:
European Commission's Horizon 2020 Research and Innovation Programme: 634495.
DOC INIA-CCAA program.
Ministerio de Economía y Competitividad de España and the Conselleria d'Educació, Cultura i Universitats Comunidad Autónoma de las Islas Baleares.
Spanish Ministry of Education, Culture and Sports (MECD): FPU13/01440.

### Competing Interests

The authors declare that they have no competing interests.

### Author Contributions

- Gaetano Catanese performed the experiments, analyzed the data, prepared figures and/or tables, authored or reviewed drafts of the paper, approved the final draft.
- Hilmar Hinz performed the experiments, analyzed the data, prepared figures and/or tables, authored or reviewed drafts of the paper, approved the final draft.
- Maria del Mar Gil performed the experiments, analyzed the data, prepared figures and/or tables, authored or reviewed drafts of the paper, approved the final draft.
- Miquel Palmer conceived and designed the experiments, analyzed the data, prepared figures and/or tables, authored or reviewed drafts of the paper, approved the final draft.
- Michael Breen analyzed the data, prepared figures and/or tables, authored or reviewed drafts of the paper, approved the final draft.
- Antoni Mira performed the experiments.
- Elena Pastor conceived and designed the experiments.
- Amalia Grau performed the experiments.
- Andrea Campos-Candela performed the experiments.
- Elka Koleva performed the experiments.
- Antoni Maria Grau conceived and designed the experiments, authored or reviewed drafts of the paper, approved the final draft.
- Beatriz Morales-Nin conceived and designed the experiments, authored or reviewed drafts of the paper, approved the final draft.

## Field Study Permissions

The following information was supplied relating to field study approvals (i.e., approving body and any reference numbers):

The collection of juvenile lobster specimens for the survival test was carried out under specific permission from Govern de les Illes Balears.

## Data Availability

The raw data are provided in the Supplemental Files.

## Supplemental Information

Supplemental information for this article can be found online at http://dx.doi.org/10.7717/peerj.4707#supplemental-information.

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
