# Peer review of "Comparing the catch composition, profitability and discard survival from different trammel net designs targeting common spiny lobster (Palinurus elephas) in a Mediterranean fishery"

_PeerJ, doi:10.7717/peerj.4707_

## Round 0.1 · original submission · Major Revisions

Reviewers' comments on your work have now been received. You will see that they suggest major revision. Please, consider all the suggestions in the revised version. The English has also to be deeply improved as well as the focus of your work in order to increase its interest.

Reviewer 1 ·

Basic reporting

The manuscipt “Comparing the catch composition, profitability and discard survival from different trammel net designs targeting common spiny lobster (Palinurus elephas) in the Mediterranean” is, as its title indicates, an attempt to evaluate the common spiny lobster fishery of the Balearic islands regarding its catches and discards, taking into account both the biological and the economic component. It is an interesting contribution providing essential knowledge for fisheries management. The article is within the PeerJ scope. The language is, generally, clear, unambiguous and professional (but see some minor corrections). The Introduction & background provide the context of the study. The literature is generally well referenced & relevant. The structure conforms to PeerJ standards (however the abstract is not a structured abstract according to the instructions). The figures are relevant, high quality, well-labelled & described (but see a comment on Figure 5, listed in detailed comments). Raw data are also supplied. Overall, I believe it will make a good contribution to the journal if accepted.

Experimental design

There is original primary research within the Aims and Scope of the journal. The research question is well defined, relevant & meaningful. It is stated how research fills an identified knowledge gap.

There might be an issue with the sampling design (see my major concern No 1), as I am not sure that the sampling scheme (covering spring and summer) employed is representative of the actual fishery. However, in the Introduction, there I a sentence that hints that this fishery is indeed carried out in these two seasons, but it seems incomplete. See my detailed comments (see Major concerns-comment No 1) below.

The methods are not described with sufficient detail & information to replicate and I believe the readers would need a bit more information (see Significant issues-comment No 3).

Validity of the findings

Generally the statistical methodologies used are adequate. I have a serious reservation about the level at which the data are analyzed (level of “netting panel” or “netting wall”) as I believe that this is problematic (see Major concerns-comment No 2). I would advise analysis at a different level (net type per fishing day/operation) to guarantee independence of observations.

Additional comments

Major concerns:
1) The authors state (Materials & Methods l. 113-114) that sampling was carried out from April to August in 2015-2016. Is this because the specific fishery for lobster is seasonal and occurring only in spring-summer months (it is written in the introduction –l. 71- but the sentence there seems incomplete and there is no further description of the fishery, see my comment No 3…)? If the answer is positive then there is no problem with the temporal aspect of the sampling design (as the sampling is representative of the period of the year of this fishing métier); however this should be clarified in the Introduction. If no, then the design of the study if flawed and I’m afraid that the manuscript should be rejected. This needs to be clarified by the authors.

2) Another important issue regards the analysis at the level of netting walls. As the authors state (l. 118-119) “A trammel net set consisted of several (between 10 and 30) netting walls (or “netting panels”)”. If I understand correctly the analysis (and the presentation of the results) is at the level of netting walls, however –since- a trammel net consists of several netting walls tied together, the observations (the catches per netting wall or the revenues per netting wall) are not independent. So, I believe that the statistical analysis should be instead carried out at the level of net-type per fishing day (the production should again be standardized per 100 m of net/per netting wall, but the catch quantity of all netting walls with similar characteristics for a given fishing day should be pooled before that standardization) and not netting wall per day. I realize that the sampling size is much lower if net types are considered and not netting walls (and it is true that the sampling effort expended is adequate, I realize it is not so easy to carry out this work on many more fishing trips), still I believe the analysis should be carried out on that level. It is OK to present catch abundance standardized per netting wall (i.e. per 100 m of net), but statistical analysis treating netting walls deployed on the same day, by the same vessel at the same location (as one net) is hinting at pseudo-replication.

Significant issues (in order of importance):
3) In general I believe it is essential to see more on the technical characteristics and a description of this fishery. This regards both (a) the Introduction part (to put this fishery into context, e.g. what are the seasons and the depths of the fishery, the common mesh sizes and the fishing effort (net length) used? The authors cite some references, but one or two sentences would be helpful for the reader) and (b) the Methodological part, so that the gears used and the locations fished are adequately described for the reader. Please be reminded that other scientists might, ideally, want to follow your experimental design to ratify your results.

4) What about the differences between the three trammel net types examined on the catches of the target species (common spiny lobster)? Some more information would be necessary to evaluate these gears.

5) Title: I believe the title of the study should be changed. As it stands, “Comparing the catch composition, profitability and discard survival from different trammel net designs targeting common spiny lobster (Palinurus elephas) in the Mediterranean” may mislead the reader to believe it examines the lobster fisheries from the entire Mediterranean. I would change it to either “Comparing the catch composition, profitability and discard survival from different trammel net designs targeting common spiny lobster (Palinurus elephas) in a Mediterranean fishery” or “Comparing the catch composition, profitability and discard survival from different trammel net designs targeting common spiny lobster (Palinurus elephas) in the Balearic islands”

6) l. 139-140: Here the reader can’t help wondering whether it is proper to assign specimens into 10-cm length classes. For some of the species caught a length class of 10 cm is not adequate as it is too broad covering a wide part of the range of between individuals variance. Furthermore, I can’t imagine which length (from this entire range) was used for the weight estimations coming from the length-weight relationships.

7) l. 119: A mesh-size of 160mm seems to be very large, even if this is the diagonal and not the knot-to-knot mesh size (still, clarify which of the two applies). Perhaps this is the size of the two wide-mesh nets (the “further nets on either side” of l. 422) and not the central (main) part? In other areas of the Mediterranean similar fisheries are carried out using old/timeworn trammel nets of various mesh sizes in the main part. Please clarify the actual mesh size of the thin mesh or whether this is the practice followed here, too (see also my general comment on the technical characteristics and description of the fishery).

8) I think that the section “Representativeness of the sampled boats” (first section of Results) would be more adequately placed in the methods and not in the results as it is the means to assure that what you did in terms of sampling was valid and representative of the entire fleet.

9) l. 334-336: The authors write “the differences in the probability of obtaining some commercial catch were relevant: MMF = 0.33 (0.23 to 0.44) versus MMF + greca = 0.44 (0.33 to 0.56)”. In l. 226-228 it was stated that “effects attributable to a given treatment
were labelled statistically not relevant or statistically relevant depending on whether their
corresponding 95% Bayesian credibility intervals overlapped or not.”, so I can’t understand how the difference is relevant since the numbers overlap.

10) l. 468-480: I think the authors should be more reluctant in generalizing their findings as the survival of lobsters didn’t include a very large sample size. The data are valuable, for sure, still the short term survival was assessed only for sixteen lobsters.

11) Figure 5: I am not sure this Figure is very clear (e.g. axis-labels).

12) l. 268: Please correct the Table order and numbers (as the numbering should be according to order of reference in the text). You are referring to Table 2 here, only to refer to Table 1 in l. 378.

13) l. 65-66: Still if you search for works on Mediterranean small-scale fisheries disards, a number of papers pop up. It would be interesting to discuss your findings in comparison to the findings of these works. This would also put the paper into context for a wider international audience in order to attract more interest.

14) Keywords: In my opinion “guarding net (greca)” and “high survival” are not very practical as keywords (especially the first one might never be searched as such). I’d advise to put “guarding net” and “discard survival”.

15) l. 227: as the “statistically relevant/not relevant” is a bit alien as an expression to the readers unfamiliar with Bayesian statistics, perhaps it would be better to intonate that by changing the sentence to “Thereafter, according to a Bayesian framework, effects attributable to a given treatment were labelled statistically not relevant or statistically relevant (contrary to “statistically non-significant”/”statistically significant” of classic inferential statistics), depending on whether their corresponding 95% Bayesian credibility intervals overlapped or not.”.

16) l. 300: Why do you consider the number of commercial individuals low? Is there any reference to compare? As you haven’t provided information on the average number of fishing netting walls for a typical fishing trip in the area in the Introduction, the reader could assume anything, I mean, isn’t it usual to put many fishing walls in a net set?

Elements of minor importance (not enumerated):
Introduction:
l. 54: change to “(Kelleher 2005)”
l. 70-73: Something seems to be missing from these sentences. Please rephrase.
l. 102: Some more information might be useful on this to put it into context. Is this practice a new initiative? Is it a common practice in this fishery?
l. 105: change to: “and to provide a detailed description”
l. 106: change to: “Additionally, we investigate”
l. 106: change to: “estimate the “short-term” (7-day)”

Materials and Methods
l. 148: change to “The quantification of this part of the catch”
l. 165: change “3” to “three”
l. 175: I believe the proper citation style isn’t by giving the website but (R Development Core Team 2008) and in the list of references as: R Development Core Team (2008) R: A language and environment for statistical computing. R Foundation for Statistical Computing, Vienna.
l. 196-198: I believe this should go before (and not after) the previous sentence.
l. 200: change to: “10-cm length classes (see justification above).”
l. 208: I am not sure this formula displays correctly on my pdf. E.g. I am not familiar with the inverted “?”. Is the formula correct? Perhaps it is the mathematic symbol for integral? Please check it out.
l. 253: change to: “The following taxa” as Raja sp. (referred to in the following line) is not a species but a genus. Except if you are referring to a specific species (though inidentified), but then this should be indicated in the text.
l. 270: change to “of the seven days”

Results:
l. 305-306 and Figures 3 and 4a: Perhaps the figures could be combined in one showing the fraction of each species that is commercial and discarded (e.g. using different coloured bars for the retained/discarded fraction).

Discussion:
l. 403: change to: “In that case”
l. 407: change to “has not been evaluated here”
l. 424: change to: “when it tries to escape”
l. 447: change to “(Kelleher 2005)”

Reviewer 2 ·

Basic reporting

• Writing is generally good, but the manuscript is overly long. The Methods could be reduced if some of the minute statistical details were placed in supplemental materials and, if as detailed below, the trivial comparison between net materials were omitted.
• The Introduction lacks a broader setting on the worldwide problem of fishing bycatch and that associated literature.
• line 86: awkward sentence structure
• line 208: the symbol " ¿" in the model is not defined.
• line 334: awkward sentence structure; do not say that something was statistically "relevant", instead indicate whether significant or not.
• line 354: missing words; should be: "...mean weight of discards was statistically different..."
• line 418: awkward sentence
• Figure 1 could be improved were labeled showing the different mesh sizes and dimensions; this goes for the greca inset figure too.
• The authors use scientific names of species throughout but many readers may be unfamiliar with these. I recommend creation of a table that lists scientific names, common names, and broad taxonomic classification of each (i.e., decapods, tunicates, etc.).
• Figure 7: graphics not described adequately in the legend.
• Too many figures: Figure 8 & 11 in particular are unnecessary.
• Table 2 unnecessary

Experimental design

• The research project is well-designed and implemented, but its relevancy is rather narrow. The great majority of spiny lobster fisheries worldwide are based on the use of traps or artificial structures referred to as "casitas", and are not net fisheries. Net fisheries for lobsters still persist primarily in developing countries with poorly managed fisheries because they are so destructive.
• The comparison between the two types of netting materials is trivial. There is no reason to suspect that nets of similar design but different materials would effect catch or by catch results; the comparison is only relevant with respect to net longevity and cost. The greta comparison is of a net of a different design, so more reasonable. The manuscript could be shortened significantly if the trivial comparison between net materials is omitted.
• The authors focus their statistical analysis on Bayesian statistics, which I am not very familiar with and so can not comment of the validity of their application.
• The sample size for the juvenile lobster post-catch survival estimation (n = 16) is low. Why were lobsters held for only 1 week? Is there some prior evidence that this is an appropriate time-frame within which one can determine bycatch discard mortality?

Validity of the findings

• Though not unusual, it is still disheartening to see that the net bycatch was 2.5 times more speciose than that of the target species and of equal biomass (means = 0.061 - 0.68 kg/net wall). Moreover, the onboard survival of most species pulled up in the nets was abysmal (0.2-0.6 probability) and probably an underestimate as many more would surely have died after returned to the sea, which was not examined except for the juvenile spiny lobster that are typically hardier than fishes.

• The authors come across as being a biased in favor of the use of this gear type, which is in appropriate. Indeed, on line 488 they state: "Based on the results reported here, it is recommended that further research should be conducted in support of an exemption from the Landing Obligation for the spiny lobster ...". Research needs to be objective and should not be conducted "in support of" one result or another. Along those same lines, the authors mention (paragraph starting at line 433) an alternative gear type (traps) and cite other studies showing that traps catch less bycatch but also fewer lobsters than nets. Yet they appear to brush off that alternative because fishers consider them less economically viable. But the data clearly show that traps cause less environmental harm than nets.

Reviewer 3 ·

Basic reporting

I am really excited to see a study that address net design and lobster bycatch/discards. Most lobster catch studies focus on trap designs, that, although comprise the bulk of lobster landings around the world, are much more studied than those smaller (but important) lobster fisheries that utilize nets. This is especially true in the Mediterranean where both spiny and slipper lobster fishery data is substantially less known compared with other areas.

This manuscript has two issues that make it difficult to grasp the important points without distractions. First, I believe the length of both the introduction and discussion sections could be shortened to eliminate redundancy and improve clarity. Second, the English grammar, text, word choice, and clarity should be amended to an international audience. I believe these two items would greatly add value, clarity, and comprehension to the science and the associated data that follows.

For both of these suggestions, I have made some example comments that are found in the PDF version of the manuscript.

Most of those graphic outputs in the figures are clear and professionally presented. I have some comments and suggestions with some of the figures:

Fig. 1: Add dimensions to the nets either in the figure caption or on the figure drawing itself. Also, I can’t really discern the difference between the greca and non-greca panel. In your illustration, it’s not entirely clear. Can you improve upon this?

Fig. 4A (and others like it): On your axis, ‘Items/netting wall’, is this given as a proportion? If so, why not a proportion out of 100? In other words, 0 to 1.0.

Fig. 8: Was there a statistically significant difference here? I think a simple 1-way ANOVA would get at this and be helpful to report.

Fig. 10: I am not entirely sure that this figure is needed at all and can be eliminated, especially with the description in your results section and the inclusion of Fig. 11.

Fig. 11: Why would your initial survival not start at 1 or 100%. The authors need to clarify this.

Raw Data: I have taken a look at the raw data files in the supplemental folder and it seems fine. I wonder if the authors ever considered placing their data into a relational database to better examine relationships.

Overall, the structure of this manuscript conforms to the ‘guidelines to authors’ instructions and an appropriate number of references appear to be represented here.

Experimental design

I believe the research is unique and original to the question(s) that are sought in this study. For example, there is a lot of attention to the design of the trammel net design and selection of the netting fiber types (very innovative and detail-oriented). The research question(s) are apparent but again, as mentioned above, could be clarified in the introduction section. There appeared to be many ‘constraints’ as the authors refer to several times. Given this, I am wondering how much confidence we can put into the subsequent results with respect to how robust they might be? With this in-mind, I have summarized some of my concerns below:

Line 208: Your model notation is confusing and difficult to decipher in the current format. I suggest changing this.

Line 165: ‘Due to logistical constrains, only 3 vessels were sampled’ – that is a really small sample size. I am wondering if you had the capacity to port sample? This is a very common practice in fisheries science.

Lines 260-278: lab study and survivorship with lobsters – I appreciate here what the authors are trying to accomplish, but with such a small sample size (n=16), it is very difficult to say anything for sure. Was there opportunities to sample other juvenile lobsters in the survey? Also, were the lobsters kept communally in a tank (this is what seems like the case) or in separate containers or aquaria tanks in the lab. The second design is much more robust and does not contribute to pseudoreplication or artifact effects.

Validity of the findings

I am not sure about the small number of replicate sampling vessels (n=3) as well as the numbers of lobsters that were used in the lab-based survivorship trial (n=16).

Lines 249-250: ‘The time that each animal had been retained in the net was unknown but was less than the maximum soaking time (48 hours)’. This could be a really wide range. Did you do some preliminary work to determine survivorship at incremental times up to 48 hours? Something like 12, 24, 36, and 48? I think this is a real gap in your data.

I appreciate the time and effort the group put into their data syntheses and analyses, especially R-statistics, and PCA outputs.

There is a good and meaningful discussion of impact and linking the original research questions to supporting results and conclusions; however, as stated previously, the Discussion section could be tailored down to remove redundancy and provide more efficient and clarified sentences.

Additional comments

A couple of other things to consider for this manuscript:

1) The title of the study is lobster specific, however a large amount of data and reporting deals with fish. I am concerned that in this case, the title is a bit misleading and doesn’t fully represent the bulk of the analyses and reporting. Please consider changing the title.
2) The title of this manuscript also implies an economic assessment (profitability) for targeting lobster, but it is not apparent to me that such an analysis was conducted in this study other than the recommendations of the authors to provide exemption of undersized lobsters from the EU ‘landing obligation’ regulation.

Annotated reviews are not available for download in order to protect the identity of reviewers who chose to remain anonymous.

---

## Round 0.2 · Minor Revisions

According to the reviewers comments and suggestions, the manuscript has to be improved. The English has to be improved.

Reviewer 1 ·

Basic reporting

The authors of the manuscript “Comparing the catch composition, profitability and discard survival from different trammel net designs targeting common spiny lobster (Palinurus elephas) in a Mediterranean fishery” have made a serious effort to change the manuscript conforming to the reviewers’ suggestions. As a result, the contribution is significantly improved and can make a good publication in PeerJ. I do not have any major objections or comments. There are still some grammar/syntax mistakes or typos that need correction and overall the English language should be improved in order to ensure that the text is understood by an international audience. Examples of the minor corrections needed are given below.

l. 81: Change “which coincides approximately with the size at first maturity” with “which approximately coincides with the size at first maturity”

l. 86, 89, 91: Change “in the Balearic Island” to “in the Balearic Islands” if you are referring to the Balearic Islands in general or to “in this Balearic Island” in the case you are referring to Majorca in particular.

l. 126: Change “data capture” to “data collection”

l. 135: Change “which is thought to reduce” to “which is considered to reduce”

l. 157: Change “that have entangled” to “that have been entangled”

l. 159: Change “of caught items” to “of entangled individuals”

l. 166: Change ““where dead” animals were observed” to “where “dead” animals were observed”

l. 169-171: Change these two sentences to “During net retrieval, small invertebrates or bed-forming organisms (e.g. Posidonia oceanica) were passively disentangling and falling on the deck in a continuous pattern; therefore, it was not possible to assign this fraction of the catch to a specific netting wall.”

l. 181: Change: “because the fishermen organize fish sales on a single fishing wharf” to “because all sales of fish are carried out in the single fishing wharf of the island”.

l. 217: Change “Due to logistical constraint” to “Due to the logistical constraint”

l. 275: Change “From above analysis,” to “In the analytical setup described above,”

l. 276-277: Change “the data was tested” to “the data were tested”

l. 345: Change “elasmobranches” to “elasmobranchs”

l. 387: Change “Thus, MMF + greca netting walls tend to retain” to “Thus, MMF + greca netting walls tended to retain”.

l. 406: Change “prob. = 0.25” to “p = 0.25”

l. 410: Change “from 8 species” to “from eight species”

l. 435: Change “is probably case specific” to “is probably case-specific”

l.446-448: As the Discussion should refer to the present tense (what you have found in the Results is now a fact and you’re discussing it) change to: “The results reported here for the greca indicate that it affects the capture of some non-commercialized species. It reduces the capture of rays but increases the capture of other smaller elasmobranch species within our study.”

l. 470: Change “our results indicated” to “our results indicate”

l. 512: Change “although base on a” to “although based on a”

l. 515: Change “be indicative for” to “be indicative of”

l. 540: Change “Another trade off” to “Another trade-off”

l. 541-546: Similar to my above suggestion, I’d change these sentences to refer to the present tense.

Experimental design

no comment

Validity of the findings

no comment

Reviewer 3 ·

Basic reporting

I think sufficient improvements have been made based on my comments and suggested changes.

Experimental design

I think sufficient improvements have been made based on my comments and suggested changes.

Validity of the findings

I think sufficient improvements have been made based on my comments and suggested changes.

Additional comments

I think sufficient improvements have been made based on my comments and suggested changes, and I appreciate the attention to the details rebuttal comments, clear changes in the document using 'Track Changes', and the cautionary approach to interpreting some of the results.

The attached Word document contains some minor corrections, spelling, grammar, etc. that I have made to this document.

Annotated reviews are not available for download in order to protect the identity of reviewers who chose to remain anonymous.

---

## Round 0.3 · accepted · Accept

We hope you continue giving us the opportunity to publish your results.

#